# The Crosstalk between Acetylation and Phosphorylation: Emerging New Roles for HDAC Inhibitors in the Heart

**DOI:** 10.3390/ijms20010102

**Published:** 2018-12-28

**Authors:** Justine Habibian, Bradley S. Ferguson

**Affiliations:** 1Cellular and Molecular Biology, University of Nevada, Reno, NV 89557, USA; jhabibian@nevada.unr.edu; 2Department of Nutrition, University of Nevada, Reno, NV 89557, USA; 3Center for Cardiovascular Research, University of Nevada, Reno, NV 89557, USA

**Keywords:** HDACs, histone deacetylases, PTMs, post-translational modifications, acetylation, lysine acetylation, heart failure, cardiac dysfunction

## Abstract

Approximately five million United States (U.S.) adults are diagnosed with heart failure (HF), with eight million U.S. adults projected to suffer from HF by 2030. With five-year mortality rates following HF diagnosis approximating 50%, novel therapeutic treatments are needed for HF patients. Pre-clinical animal models of HF have highlighted histone deacetylase (HDAC) inhibitors as efficacious therapeutics that can stop and potentially reverse cardiac remodeling and dysfunction linked with HF development. HDACs remove acetyl groups from nucleosomal histones, altering DNA-histone protein electrostatic interactions in the regulation of gene expression. However, HDACs also remove acetyl groups from non-histone proteins in various tissues. Changes in histone and non-histone protein acetylation plays a key role in protein structure and function that can alter other post translational modifications (PTMs), including protein phosphorylation. Protein phosphorylation is a well described PTM that is important for cardiac signal transduction, protein activity and gene expression, yet the functional role for acetylation-phosphorylation cross-talk in the myocardium remains less clear. This review will focus on the regulation and function for acetylation-phosphorylation cross-talk in the heart, with a focus on the role for HDACs and HDAC inhibitors as regulators of acetyl-phosphorylation cross-talk in the control of cardiac function.

## 1. Introduction

Heart failure (HF) is a common condition in the United States that impacts over five million Americans. Moreover, eight hundred thousand individuals are diagnosed with HF annually [1,2]. HF is a clinical syndrome that is defined by structural and functional defects in the myocardium, resulting in the impairment of ventricular filling (i.e., diastole) or ejection of blood (i.e., systole) [3]. Diagnosis of HF is often accompanied by a collection of signs and symptoms, such as: shortness of breath or orthopnea upon lying down; edema; fatigue, weakness, or lethargy; abdominal distention; right hypochondrial pain; and/or, paroxysmal nocturnal dyspnea [3]. As a result, HF can be classified as either acute or chronic and its etiology is due to a variety of factors. The most common clinical manifestation includes reduced left ventricular myocardial function [3]. However, other causes include dysfunction of the pericardium, myocardium, endocardium, heart valves, and/or great vessels, cardiac fibrosis, scar formation, and loss of cardiomyocytes [3]. HF often requires hospitalization and it is associated with a 50% five-year survival rate; prognosis for individuals who are diagnosed has remained poor over the past twenty years [1,2]. Treatment usually requires lifelong management and it is centered on symptom management via utilization of medications, dietary changes, and reduction in hospital stay. Medication management includes preventing the collection of water within the extracellular components with the use of diuretics [1,2], or inhibition of signaling pathways on cell receptor sites with the use of angiotensin converting enzyme inhibitors and beta-adrenergic blockers [4]. Other medications include aldosterone antagonists, digoxin, anticoagulants, and inotropic agents [1,2]. However, these mechanisms for symptom management, listed above, have not resulted in marked improvements in five-year mortality rates. Therefore, there is an urgent need for improved therapeutics that have the potential to halt and/or reverse the structural and functional defects in the myocardium that lead to HF.

Substantial evidence highlights histone deacetylases (HDACs) as an intracellular therapeutic target for the treatment of HF [5,6,7,8,9,10,11,12]. Historically, HDACs were studied as regulators of nucleosomal chromatin in which they altered gene expression patterns by targeting transcriptional activity [5,9,13]. However, recent evidence suggests that the deacetylation of histone and non-histone proteins impacts other post-translational modifications (PTMs) that control intracellular signaling and gene expression [14,15,16,17]. This would suggest that treatment with HDAC inhibitors not only regulates gene expression via canonical control of DNA accessibility, but also impacts PTM cross-talk. This review will discuss the role for HDACs in the regulation of acetylation-phosphorylation cross-talk, with an emphasis on HDAC inhibition as a regulator of acetyl-controlled protein phosphorylation in the treatment of cardiac disease. 

## 2. Histone Deacetylases (HDACs)

Nucleosomes, the structural units of chromatin, are composed of DNA and histone proteins, which are essential for DNA packaging in eukaryotic cells [18,19,20]. The nucleosome consists of an octamer that is an H3-H4 tetramer and two H2A-H2B dimers in which DNA is wrapped around the octamer [18,19,20]. Lysine acetylation of these histone tails by histone acetyl transferases (HATs) results in the relaxation of the chromatin structure, creating an environment for improved transcriptional activation [20,21]. HDACs function in catalyzing the removal of an acetyl group from lysine residues on histone tails. Therefore, when HDACs are more active, histone proteins are bound more tightly to DNA, making it difficult for transcriptional proteins to combine with DNA, resulting in the inhibition of gene transcription [20,21].

The role of HDACs in the development and progression of disease, specifically cardiac disease, has often been observed since histone acetylation is a key component in the regulation of gene expression [6,12,16,22,23,24,25,26,27]. There are eighteen known mammalian HDACs that are categorized into four classes: class I HDACs (1, 2, 3, and 8), class II (4, 5, 6, 7, 9, and 10), class III, which are sirtuins (Sirt 1-7), and the lone class IV HDAC (HDAC11) [13,21]. Class I, II, and IV HDACs are zinc-dependent enzymes. Class I HDACs are present in all cells, and they were traditionally considered nuclear histone deacetylases. Class II HDACs have increased expression in striated muscle and the brain with an extended N-terminus and catalytic domain [13,21]. Moreover, class II HDACs can be further divided into class IIa and class IIb HDACs. Class IIa HDACs (4, 5, 7, and 9) carry the ability to shuttle between the nucleus and the cytoplasm and are expressed in the brain, heart, and muscle [28]. Class IIb HDACs (6 and 10) are primarily located in the cytoplasm [28], although HDAC6 has also been reported to localize to the sarcomere [7]. Class III HDACs or sirtuins constitute a subfamily of NAD+ dependent enzymes to catalyze its activity [13,21]. Finally, the sole class IV HDAC, HDAC11, shares sequence homology with class I and class II HDACs [13,21]. The function of HDAC11 is in protein stability of DNA replication factor CDT1 and IL10 [29], as well as transcriptional regulation when complexed with bromodomain protein 2 [30]. 

The connection between HDACs and the heart was initially observed by the discovery that class IIa HDACs interact with the transcription factor, myocyte enhancer factor-2 (MEF2) [31,32,33,34,35,36,37,38]. MEF2 is a key regulator of cardiac hypertrophy [39,40,41]. Overexpression of class IIa HDACs 4, 5, or 9 suppressed MEF2 activity, while the knockout of HDAC9 resulted in profound cardiac hypertrophy and dysfunction [32,34,35,38], it was hypothesized that HDAC inhibition would promote cardiac dysfunction and disease. However, HDAC inhibition blocked cardiac hypertrophy [27], suggesting that distinct HDACs promote cardiac hypertrophy, while other HDACs inhibit cardiac hypertrophy. It is now known that pan-HDAC inhibitors, which were used in these early studies, specifically targeted class I and IIb HDACs with limited inhibition of class IIa HDACs [6,9]. Moreover, evidence suggests that class IIa HDACs regulate transcriptional repression through their ability to recruit corepressor complexes to target genes, while class I HDACs act as epigenetic regulators of nucleosomal chromatin deacetylation [6,16,27,32,42]. Since these early studies, mounting evidence demonstrates that HDAC inhibitors are efficacious in models of pathological cardiac remodeling and heart failure, yet the molecular mechanisms of HDAC actions are continually being explored. Furthermore, the beneficial effects of HDAC inhibitors on heart failure and cardiac function have been attributed to the inhibition of zinc-dependent HDACs [6,12,16,25,26,43]. Therefore, this review will focus on zinc-dependent HDACs. 

## 3. HDAC Inhibitors in the Heart

As mentioned above, HDACs catalyze the removal of acetyl groups from lysine residues on histone proteins and they were historically studied as epigenetic regulators of gene transcription [21]. HDACs are zinc metal hydrolases and HDAC inhibitors are zinc ion chelators. HDAC inhibitors are comprised of a “cap” structure that interacts with amino acids located at the entrance of an *N*-lysine binding channel and a “chelator” that binds a zinc ion present in the HDAC enzyme active site. This is “linked” to the cap structure by a five or six hydrocarbon chain [4,13,44]. The cap and chelator direct isoform selectivity; five different classes of HDAC inhibitors interact in this pharmacophore model [45].

HDAC inhibitors are efficacious in animal models of cardiac remodeling and heart failure [6,7,8,10,11,14,22,46,47,48,49]. Cardiac fibroblasts are the most abundant cells in mammalian heart tissue as they provide structure to the heart [50], yet following cardiac injury, e.g., a myocardial infarction, cardiac fibroblasts transdifferentiate into active myofibroblasts that promote collagen deposition and cardiac fibrosis [51,52]. Cardiac fibrosis contributes to cardiac remodeling and dysfunction that leads to and/or exacerbates congestive heart failure (CHF) [50], HDAC inhibitors, and in particular class I and IIb HDAC inhibitors, attenuate cardiac fibrosis and improve systolic and diastolic heart function [6,14,22,23,43,51]. The first suggestion for the role of class I HDAC inhibitors stemmed from observations that SK-7041 inhibited cardiac hypertrophy and fibrosis in response to angiotensin II or aortic banding [26]. However, it was later reported that SK-7041 acts as a pan-HDAC inhibitor [53]. Reports later confirmed that class I HDAC inhibition, using MGCD0103 (Mocetinostat), attenuated angiotensin II-induced fibrosis in rodents [6] and in CHF rats [23,43]. 

HDAC inhibition has also been shown to inhibit pathological cardiac hypertrophy and improve muscle contractility, as well as attenuate systolic and diastolic dysfunction [7,10,11,14,16,27]. More recently, HDAC inhibitors were shown to stop and potentially reverse cardiac remodeling in response to ischemia-reperfusion injury and in aged animals [11,14]. Ischemia is the prolonged blockage of coronary blood flow that can result in cardiomyocyte apoptosis and necrosis [11], despite the reperfusion or restoration of blood flow to the heart. SAHA, a pan-HDAC inhibitor, reduced infarct size and attenuated systolic dysfunction in rabbits when administered before the ischemic event or at the time of reperfusion, after ischemia, which led to the reduction in infarct size and improvements in systolic function [11]. 

In addition to systolic failure, chronic heart failure is also characterized by left ventricular diastolic dysfunction or heart failure with preserved ejection fraction (HFpEF) [14]. In this case, left ventricular relaxation is impaired, which contributes to the inadequate filling of blood, despite maintained systolic function [14]. HFpEF is common in aged population. Jeong, et al. [14], demonstrated that treatment with ITF, a pan-HDAC inhibitor, improved diastolic function in aged rodents. Combined, these data clearly demonstrate the efficacy for zinc-dependent HDAC inhibitors for the treatment of heart failure in rodent and large animal models of cardiovascular disease. However, these data also demonstrate that HDAC inhibitors improve heart function through both epigenetic and non-epigenetic mechanisms. Impacts for HDAC inhibition have been summarized in Table 1. Below, we will highlight the role for acetylation in the cross-talk and regulation of protein phosphorylation. 

## 4. Protein Phosphorylation-Acetylation

Posttranslational modification (PTM) of proteins is crucial for the modulation and regulation of many protein-protein interactions and functions [56,57]. PTMs that have been identified on histone proteins include acetylation, ubiquitination, phosphorylation, and methylation, yet PTMs occur on many proteins within the cell [57]. Cells consistently adapt to a changing environment with the assistance of these PTMs, which have a profound impact on intracellular signaling, protein-protein interaction, and the regulation of gene expression, among other cellular actions [56,58]. Protein phosphorylation is the most studied PTM, yet only recently have studies begun to highlight the role for cross-talk between phosphorylation and acetylation in the regulation of cellular fate [57].

Protein phosphorylation is a reversible PTM that is essential for protein synthesis, cellular proliferation, apoptosis, signal transduction, growth, development, and aging [59]. The human genome consists of protein kinases and phosphatases, which govern protein phosphorylation and dephosphorylation, respectively [59]. Of the approximately 500 human protein kinases, most regulate the phosphorylation of serine and threonine (serine/threonine protein kinases), tyrosine (tyrosine protein kinases), or all three residues (dual-specificity kinases) [59]. Conversely, protein phosphatases dephosphorylate (i.e., remove phosphate groups), tyrosine (protein tyrosine phosphatases), serine and threonine (serine-threonine phosphatases), or all three residues (dual-specificity phosphatases) [60,61,62]. The addition of the negatively charged phosphate group to a protein can impact protein conformation; this can have profound effects on protein activity, protein stability, protein interactions, and protein localization [59]. For example, the phosphorylation of tumor suppressor protein p53 can increase p53 protein stability, transcriptional activity, and nuclear localization, which results in the transcription of genes that inhibit the cell cycle, activates DNA repair, and, at times, induce apoptosis [63,64,65,66]. In the heart, p53 is known to regulate cardiac transcriptomic changes that govern cardiac architecture, excitation-contraction coupling, mitochondrial biogenesis, and oxidative phosphorylation [63,67]. Phosphorylation of p53 at serine 15 (Ser15) and Ser20 has been shown to stimulate growth arrest and DNA damage inducible protein 45β (Gadd45β) transcription; this drives cardiac injury in response to myocardial ischemia [64]. Combined, these data highlight the importance for p53 and p53 phosphorylation in the regulation of cardiac fate.

Similar to phosphorylation, acetylation occurs on lysine residues of histone and non-histone proteins [68,69,70,71,72,73]. Recent reports have shown that approximately 4500 proteins can be acetylated on [68] about 15,000 lysine residues in rodents [68]. These and other findings have led to the suggestion that protein acetylation rivals the kinome in cellular importance [68,71,72,74,75,76,77]. Similar to histones, HATs have been shown to govern protein acetylation, while HDACs regulate protein deacetylation of lysine residues. Like kinases and phosphatases, the extent of HAT/HDAC activity and localization controls protein acetylation duration, magnitude, and compartmentalization, which ultimately dictates protein activity, stability, and function [68,75,78,79,80,81]. Similar to phosphorylation, p53 can be acetylated [82,83,84,85]. Indeed, p53 acetylation is necessary for cellular apoptosis and growth arrest [82,83,84,85]. In these studies, acetylation of lysine residue 120 (Lys120) was crucial for p53-mediated apoptosis; this modification inhibits mouse double minute 2 (Mdm2)-mediated p53 proteasome degradation [84,85]. In addition, the attenuation of p53 acetylation was shown to inhibit p53 transcriptional activation, which is critical for p53 binding to gene regulatory elements, particularly to the promoter region of cyclin dependent inhibitor kinase 1A (p21); p53 binding increased p21 gene expression and subsequent growth arrest [82,83]. Upon stress or DNA damage, the HATs p300/CBP, PCAF, or TIP60 can acetylate p53, which results in p53 protein stability and binding to regulatory response elements to promote apoptosis and cell cycle arrest [82,84]. Of note, reducing p53 acetylation in the heart was shown to attenuate chemotoxicity, suggesting important roles for p53 function in the cardiac tissue of cancer patients [86]. Additionally, p53 acetylation has been implicated in myocardial infarction, in which deacetylation of Lys118 on p53 is necessary for nitric oxide synthase 3 (NOS3) transcription and cardioprotection in the infarcted heart [87]. Thus, these findings demonstrate that p53 acetylation switches p53 from pro-survival to pro-apoptotic in the heart. 

Given the multitude of modification sites, p53 activity and stability are likely redundantly regulated by many PTMs. Moreover, multiple modification sites on p53 suggests cross-talk between PTMs in the regulation of p53 function. This is evident from studies that report that p53 phosphorylation of Ser15 and Ser46 are necessary for p53 acetylation [65,66]. Moreover, this phosphorylation-acetylation cross-talk regulates cellular fate, as p53 phosphorylation of Ser46 leads to the acetylation of Lys382; Lys382 is necessary for p53-mediated apoptosis [65]. In response to DNA damage, p53 phosphorylation promotes p300 HAT binding to p53, which contributes to p53 acetylation [66,88]. Four HDACs (HDAC1, HDAC3, SIRT1, and SIRT7) have been shown to deacetylate p53 [89,90,91,92]. Together, these findings suggest that cross-talk between phosphorylation and acetylation potentially impacts cardiac function. Indeed, Ser15 phosphorylation stimulates p53 acetylation and both p53 acetylation and phosphorylation, particularly at Ser15 [64], drives cardiac injury. Thus, therapeutic inhibition of one or both of these PTMs may improve cardiac function. However, less is known regarding how acetylation functions to regulate phosphorylation. Below, we will highlight new findings for the role of acetylation in the regulation of protein phosphorylation. 

## 5. HDAC Inhibition, Protein Phosphorylation, and Heart Failure

As discussed above, HDAC inhibitors are efficacious in pre-clinical animal models of HF [6,7,8,11,14,23,43,49]. However, molecular underpinnings for HDAC inhibitors in the heart have largely focused on their role in the regulation of histone (de)acetylation and gene expression [47,93]. Recent reports, however, have shown that HDAC inhibitors can target cytosolic and sarcomere proteins to improve cardiac function via improvements in muscle contractility [7], relaxation [14], autophagy [11], and proteotoxicity [8]. These epigenetic and non-epigenetic roles for HDACs and HDAC inhibitors suggest multiple mechanisms for cardiac protection. 

### 5.1. HDAC Inhibition and Protein Dephosphorylation

As phosphorylation has been shown to impact protein acetylation [65,66,88], it should be no surprise that acetylation can, in turn, regulate protein phosphorylation. Early studies linked cross-talk between acetylation and phosphorylation through canonical regulation of histone acetylation and changes in gene expression. Indeed, reports in the heart demonstrated that class I HDACs repressed phosphatase gene expression in response to stress, and that treatment with class I HDAC inhibitors altered histone acetylation, leading to the induction of dual-specificity phosphatase 5 (dusp5) and the subsequent dephosphorylation of extracellular signal-regulated kinase 1/2 (ERK1/2) [16]. In this report, class I HDAC inhibitors attenuated stress-induced cardiac hypertrophy, in part through phosphatase-mediated dephosphorylation of nuclear ERK1/2 [16]. Similar reports showed that inhibition of HDAC3 attenuated diabetic cardiomyopathy (DCM). In response to DCM, ERK1/2 phosphorylation is increased concomitant to decreased histone acetylation of the dusp5 gene and repression of dusp5 [46]. Treatment with class I HDAC inhibitors or inhibitors that are specific for HDAC3 led to increased histone H3 acetylation at the dusp5 gene promoter, re-expression of the dusp5 gene, and inhibition of ERK1/2 phosphorylation [46]. These reports suggest that HDAC inhibitors work via canonical regulation of histone acetylation that leads to inducible phosphatase gene expression in the feedback control of protein phosphorylation in the heart.

Phosphatases can be regulated by gene expression or PTMs that control phosphatase protein stability and activity [59,94]. Indeed, the dual-specificity phosphatase 1 (dusp1) has been shown to be acetylated by the HAT p300 on Lys57 [95]. Acetylation of dusp1 was shown to enhance dusp1 binding to p38, increasing dusp1 phosphatase activity, and dephosphorylating p38 MAPK [95]. Moreover, treatment with pan- or class I HDAC inhibitors increased dusp1 acetylation, resulting in p38 dephosphorylation and the attenuation of inflammation in vivo. These effects were dependent on dusp1 acetylation as anti-inflammatory impacts for HDAC inhibitors were lost in dusp1 knockout mice [95]. These findings are interesting as p38 MAPK phosphorylation has also been implicated in pathological cardiac hypertrophy and cardiac dysfunction [96,97,98,99]. Thus, these findings would suggest that HDAC inhibitors in the heart potentially activate dusp1, without changing gene expression, to dephosphorylate p38 in order to attenuate hypertrophic signaling and improve muscle function. Similar observations have been made with phosphatase and tensin homolog (PTEN) [55]. HDAC6 was shown to deacetylate PTEN, inhibiting PTEN translocation to the membrane, leading to increased Ser473 phosphorylation of Akt. In contrast, pharmacological or genetic inhibition of HDAC6 resulted in increased PTEN acetylation, membraned translocation, and attenuation of Akt phosphorylation [55]. Loss of PTEN in the heart has been shown to increase cardiac hypertrophy and drive contractile defects via increased Ser473 phosphorylation [100]. Moreover, genetic or pharmacological inhibition of HDAC6 has been shown to improve contractile function in the heart [7]. Thus, these data would suggest that protective actions for HDAC6 inhibitors are mediated in part through PTEN acetylation, membrane translocation, and attenuation of Akt phosphorylation.

### 5.2. HDAC Inhibition and Protein Kinases

The findings above support a role for acetylation in the regulation of protein phosphatases as feedback inhibitors of protein phosphorylation. However, direct actions have been attributed to the acetylation of protein kinases in regulating protein kinase activity [17,54,101]. For example, it was reported that p38 MAPK is acetylated on Lys152 and Lys53 [54]. Acetylation of p38 enhanced p38 activity by increasing ATP binding affinity. In response to stress, p38 acetylation is increased in cardiomyocytes as a result of increased p300 and PCAF HAT activity [54]. Conversely, HDAC3 was shown to deacetylate p38 in cardiomyocytes; this corresponded to increased p38 acetylation and phosphorylation in cardiac myocytes in response to HDAC inhibition [54]. What this means for HDAC inhibitor therapy is still unclear, as p38 activation has been implicated in pathological cardiac hypertrophy [96,97,98,99], although these studies remain inconclusive [102,103]. Similar to p38, its upstream kinase mitogen-activate protein kinase kinase 6 (MKK6) is also acetylated [101]. However, acetylation of MKK6 within the activation loop blocked phosphorylation and activation of the protein. Interestingly, MKK6 acetylation led to the inhibition of downstream targets, such as NF-κB signaling [101]. NF-κB and p38 have been shown to collaborate in cardiac myocytes to protect cells from apoptosis [104]. Under these conditions, MKK6-bound to IκB kinase beta (IKK-β), which led to IκB phosphorylation and degradation, resulting in NF-κB activation. MKK6-mediated NF-κB activation induced interleukin-6 (IL-6) gene expression in a p38-dependent manner [104]. Thus, these data would suggest that treatment with HDAC inhibitors would increase MKK6 acetylation, which would impact the MKK6- NF-κB activity and thus alter the inflammatory response in cardiomyocytes. 

Findings for these direct actions for acetylation in the regulation of protein phosphorylation are not surprising, given recent proteomic observations that other protein kinases are regulated by acetylation. In these studies, HDAC inhibition with trichostatin A (TSA) led to increased phosphorylation of protein tyrosine kinase 2 (PTK2), mitogen-activated protein kinase 3 (MAPK3; ERK1), and muscle pyruvate kinase, among others [17]. It should be noted that HDAC inhibition did not increase the phosphorylation of all protein kinases. However, these findings suggest that treatment with HDAC inhibitors can broadly impact protein phosphorylation in the cell by regulating protein kinase activity, and by result downstream protein phosphorylation. The protein kinase that is targeted by HDACs can itself have profound consequences on the cardiac transcriptome. This is evident from studies that examined acetylation of cyclin-dependent kinase 9 (CDK9) on Lys 44 and Lys 48 within the CDK9 catalytic core. CDK9 activity is essential for RNA polymerase II (RNAPII) phosphorylation, which promotes RNAPII transcriptional elongation [105,106]. GCN5 and PCAF promote CDK9 acetylation, which inhibits the phosphotransfer reaction and limits positive transcription elongation factor b (P-TEFb) kinase function and transcriptional activity [107]. These data could explain why treatment with HDAC inhibitors, which would lead to increased acetylation of CDK9, is often times marked with decreased gene expression for fetal and pathological cardiac genes that improve cardiac function [27]; this is antithetical to the dogma for HDAC-mediated gene expression. Thus, acetyl-mediated regulation of kinase activity by HDAC inhibitors can have profound impacts on the cardiac transcriptome that are mediated via non-epigenetic or non-traditional functions that are aligned to histone acetylation and DNA accessibility.

## 6. Conclusions

This review covered the crosstalk between protein phosphorylation and lysine acetylation and highlighted an emerging role for HDAC inhibitors in the regulation of acetylation-phosphorylation cross-talk for the treatment of cardiac dysfunction and failure. In this review, we discussed the roles for HDACs as well as their inhibition in the regulation of phosphorylation that was mediated through the induction of phosphatase gene expression (Figure 1A) or activation of constitutively expressed phosphatases (Figure 1B). In addition, this review discussed direct regulation for acetylation on protein kinase activity. Moreover, we discussed how these changes could impact cardiac remodeling and function, highlighting both epigenetic and non-epigenetic roles for HDAC inhibitors in the treatment of cardiac dysfunction and failure. However, we did not touch on the crosstalk between histone acetylation-phosphorylation. This too is an important topic that adds complexity to our understanding for HDAC inhibitors in the heart. Histone proteins, similar to non-histone proteins, undergo phosphorylation-acetylation events. In addition, we failed to cover the role for acetylation-phosphorylation cross-talk in the regulation of “reader” proteins, like Bromodomain-containing protein 4 (BRD4) and its impact on gene transcription (this has been studied here [108]). 

Likewise, this review specifically examined a role for zinc-dependent HDACs in the regulation of acetylation-phosphorylation crosstalk. However, class III HDACs, the sirtuins also impart acetylation changes and are important in the heart [109,110,111,112]. Sirtuins are NAD^+^-dependent HDACs, and as such, they are sensitive to changes in nutrient utilization and ATP needs, as NAD^+^/NADH ratios impact sirtuin activity. Ample studies have examine sirtuins in the heart [110,113,114,115,116], yet only a few have examined their role in the regulation of acetylation-phosphorylation cross-talk [116].

In summary, current research demonstrates pre-clinical efficacy for HDAC inhibitors in small and large animal models of HF. That, paired with the FDA approval of four HDAC inhibitors (vorinostat, romidepsin, belinostat, and panobinostat) for the treatment of cancer, suggests feasibility in repurposing HDAC inhibitors for the treatment of human HF; there are currently no ongoing clinical trials using HDAC inhibitors in HF patients. However, as this review points-out, HDAC inhibitors likely target more than just histone proteins in the myocardium. As such, we would argue the importance for considering class/isoform selective HDAC inhibitors to limit negative off-target actions that may occur with treatment from pan-HDAC inhibitors that are currently FDA approved. Consistent with this, a recent systematic review of pan-HDAC inhibitors in cancer patients demonstrated mild cardiac side effects that included ST-segment and T wave (ST/T) abnormalities along with QT prolongation; this is a concern, as these measures demonstrate the potential for lethal ventricular arrhythmias [117]. Overall, the authors noted global cardiovascular safety for HDAC inhibitors in cancer patients. It should also be noted that dosing time, drug concentration, and HDAC isoform selectivity would likely minimize concerns observed in these cancer patients. In keeping, prospective examination demonstrated reduced stroke risk in epileptic patients that were prescribed sodium valproate (SVA), a short chain fatty acid class I HDAC inhibitor, as compared to non-SVA prescribed patients or antiepileptic drug (AED) prescribed patients [118]. Moreover, SVA patients (17 of 168) had lower stroke recurrence when compared to AED patients (105 of 530) [118]. While this study is observational, it suggests that targeting class I HDACs for inhibition can improve cardiovascular disease risk. Future work in this field is needed to delineate the safety and efficacy of these compounds in human heart failure patients, as well as determine the global mechanistic actions for these drugs in the heart.

## Figures and Tables

**Figure 1 ijms-20-00102-f001:**
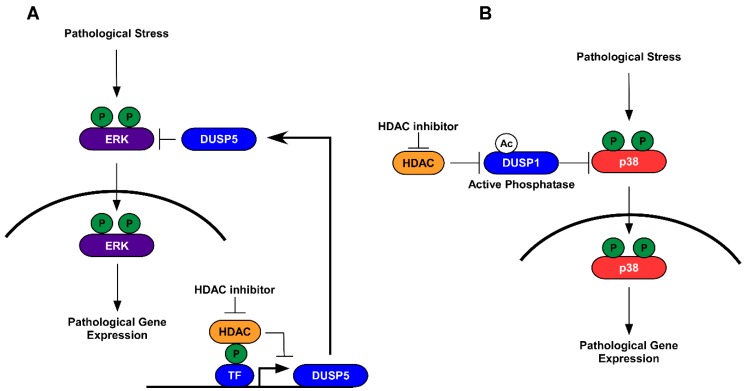
Schematic depicting the role for acetylation-phosphorylation cross-talk in the regulation of cardiac gene expression. (**A**) histone deacetylases (HDACs) regulate histone protein deacetylation that can suppress protein phosphatase gene expression. HDAC inhibition attenuates phosphatase (DUSP5) gene expression that in turn dephosphorylates extracellular signal-regulated kinase 1/2 (ERK1/2) and attenuates pathological gene expression. (**B**) HDACs regulate phosphatase deacetylation, which inactivates the phosphatase DUSP1. HDAC inhibition leads to DUSP1 acetylation and activation, which subsequently dephosphorylates p38 to inhibit pathological gene expression.

**Table 1 ijms-20-00102-t001:** Histone deacetylases (HDAC) inhibitors and their known roles in the heart.

HDAC Inhibitor	HDAC Class	Known Actions in the Heart
Scriptaid	Pan- HDAC inhibitor	Attenuated interstitial collagen deposition with angiotensin II treatment [6]
MGCD0103 (Mocetinostat)	Class I HDAC selective inhibitor	Inhibited cardiac fibrosis in response to angiotensin II [6] Attenuated fibrocyte differentiation into active fibroblasts [6] Reduced cardiac fibrosis in rats [6,23,43] Inhibited profibrotic CTGF [23,52] Attenuated ERK 1/2 phosphorylation [16]
Suberoylanilide Hydroxamic acid (SAHA)	Pan-HDAC inhibitor	Reduced infarct size and attenuated systolic dysfunction in rabbits [11]
ITF	Pan-HDAC inhibitor	Improved diastolic function in aged rodents [14]
SK-7041	Pan-HDAC inhibitor	Inhibited cardiac hypertrophy and fibrosis in response to aortic banding [26]
TSA	Class I and II HDAC inhibitor	Increased p38 phosphorylation in cardiac myocytes [54] Increased phosphorylation of PTK2, MAPK3, ERK1 [17]
RGFP966	HDAC3 (class I HDAC) inhibitor	Attenuates diabetic cardiomyopathy through histone acetylation at dusp5 gene promoter [46] Attenuated ERK1/2 phosphorylation [16]
Sulforaphane	Class I and II HDAC inhibitor	Deacetylation of PTEN increased Ser473 phosphorylation of Akt [55]
Tubastatin A	HDAC6 (Class IIb HDAC) inhibitor	Improved contractile function in the heart [7]

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
