# Peer review of "The Crosstalk between Acetylation and Phosphorylation: Emerging New Roles for HDAC Inhibitors in the Heart"

_ijms, 2018, doi:10.3390/ijms20010102_

Reviewer 1 Report

The manuscript by Habibian and Ferguson provides a comprehensive overview of the significance and emerging roles of HDAC and their inhibitors in the heart. The aspect of post-translational modifications and especially the crosstalk between phosphorylation and acetylation is very interesting. Given that there is a lot of information provided, the manuscript could benefit from some re-organization in order to make the key points clearer to the reader.  

Comments to the manuscript:

-Line 92, page2: what are Class IIb and Class IIa? They have not been defined.

-Why concentrate only on zinc-dependent HDACs? What is the rationale behind this choice?

-While the authors state that they will concentrate only on zinc-dependent HDACs, there is actually a section in the conclusion describing the impact of class III HDACs on acetylation. This section should either be removed or mentioned at an earlier point of the manuscript, not in the conclusion.

-Similarly, the paragraph in the conclusion describing BRD4 should be included at an earlier point and not in the conclusion.

-In section 3, most examples presented are pan-HDAC inhibitors. Are there any examples of class-or isoform-specific HDAC inhibitors?

-Any information that can be presented in a table or figure format would be very helpful to the reader. For example, a figure showing the PTMs of p53, their significance and changes in disease/stress conditions as described in the text, especially in page 4, would be very helpful. Also, the authors could consider including a table with a list of HDAC inhibitors and their use in different studies.

-Given that the main focus of the manuscript is the crosstalk between acetylation and phosphorylation, section 3 ‘HDAC inhibitors in the Heart’ should be condensed.

-Is there any information available on the use HDAC inhibitors in human heart disease patients?

-The authors should also briefly mention the toxicity and especially cardiotoxicity issue of HDAC inhibitors. This is an important matter that has been under investigation, especially in the cancer field. For example, there is a meta-analysis study reporting cardiac side effects in HDAC inhibitor-treated patients.

-Figure, especially panel A, is not clear. The authors could consider having a separate panel for pathological stress, so as to show the changes that occur on the acetylation-phosphorylation cross-talk upon heart disease/stress.

-Do the authors believe that the phosphorylation-acetylation crosstalk or the acetylation-phosphorylation crosstalk is more significant in the control of cardiac function?

-Are there any studies describing the use of HDAC inhibitors in heart disease animal models?

Minor comment:

Line 203, page5: deacetylase should be corrected to deacetylate?

Author Response

response: We would like to thank reviewer 1 for their helpful feedback regarding our manuscript. We have made changes as requested and feel that the quality of the manuscript has improved thanks to reviewer comments. Our changes will be highlighted in yellow in the manuscript. Please see below for our responses to the reviewer concerns.

Comments to the manuscript:

-Line 92, page2: what are Class IIb and Class IIa? They have not been defined.

response: We agree that we should have discussed this in more detail and have now added more information on page 2, highlighted in yellow, discussing class IIa and IIb HDACs.

-Why concentrate only on zinc-dependent HDACs? What is the rationale behind this choice?

response: We primarily touch on zinc-dependent HDACs over sirtuins, as inhibition of zinc-dependent HDACs is cardio-protective. However, this doesn’t mean that sirtuins are unimportant in the heart. We have added more information for our rationale of zinc-dependent HDACs, see highlighted section on page 3.

-While the authors state that they will concentrate only on zinc-dependent HDACs, there is actually a section in the conclusion describing the impact of class III HDACs on acetylation. This section should either be removed or mentioned at an earlier point of the manuscript, not in the conclusion.

response: We have removed most of this from the discussion. See yellow highlighted.

-Similarly, the paragraph in the conclusion describing BRD4 should be included at an earlier point and not in the conclusion.

response: We have removed most of this from the discussion. See yellow highlighted.

-In section 3, most examples presented are pan-HDAC inhibitors. Are there any examples of class-or isoform-specific HDAC inhibitors?

response: There are multiple reports demonstrating that inhibition of class I HDACs reduces cardiac remodeling and improves cardiac function in animals. In addition, inhibition of HDAC6 via pharmacology or genetics, improves cardiac contractility. We have made sure to discuss this in section 3. See highlighted sections. We have also added a brief section at the end of the discussion.

-Any information that can be presented in a table or figure format would be very helpful to the reader. For example, a figure showing the PTMs of p53, their significance and changes in disease/stress conditions as described in the text, especially in page 4, would be very helpful. Also, the authors could consider including a table with a list of HDAC inhibitors and their use in different studies.

responses: We have included a table with the HDAC inhibitors and known roles in the heart or known targets.

-Given that the main focus of the manuscript is the crosstalk between acetylation and phosphorylation, section 3 ‘HDAC inhibitors in the Heart’ should be condensed.

response: We have condensed this section as suggested.

-Is there any information available on the use HDAC inhibitors in human heart disease patients?

response: We have included a retrospective study in epileptic patients in our discussion. See highlighted section.

-The authors should also briefly mention the toxicity and especially cardiotoxicity issue of HDAC inhibitors. This is an important matter that has been under investigation, especially in the cancer field. For example, there is a meta-analysis study reporting cardiac side effects in HDAC inhibitor-treated patients.

response: We have included this topic in our discussion. See highlighted section.

-Figure, especially panel A, is not clear. The authors could consider having a separate panel for pathological stress, so as to show the changes that occur on the acetylation-phosphorylation cross-talk upon heart disease/stress.

response: We agree that this was not clear and have made substantial changes to the figure.

-Do the authors believe that the phosphorylation-acetylation crosstalk or the acetylation-phosphorylation crosstalk is more significant in the control of cardiac function?

response: That is a good question. I’m not sure which is fundamentally more important. However, we are implying that inhibition of HDAC activity with HDAC inhibitors will impact cell function, not just through regulation of histone acetylation, but also through changes in other PTMs.

-Are there any studies describing the use of HDAC inhibitors in heart disease animal models?

response: Yes and we have added this to the manuscript. See highlighted, page 3.

Minor comment:

Line 203, page5: deacetylase should be corrected to deacetylate?

response: We have made that change. See highlighted section.

Reviewer 2 Report

The review is focused on he possible role of histone deacetylase inhibitors in heart failure. The topic is interesting and the review is well written.

However, a paragraph about the available therapeutic approaches and the future therapeutic perspectives in heart failure should be added.

Author Response

response: We would like to thank reviewer 2 for their helpful feedback regarding our manuscript. We have made changes as requested and feel that the quality of the manuscript has improved thanks to reviewer comments. Our changes will be highlighted in yellow in the manuscript. Please see below for our responses to the reviewer concerns.

We have added more on this topic at the end of the discussion. See highlighted section.

Round  2

Reviewer 1 Report

accept